# Review of Thermochemical Technologies for Water and Energy Integration Systems: Energy Storage and Recovery

**Miguel Castro Oliveira** [1,2], **Muriel Iten** [1,*] **and Henrique A. Matos** [2]

1 Low Carbon and Resource Efficiency, R&Di, Instituto de Soldadura e Qualidade, 4415-491 Grijó, Portugal; dmoliveira@isq.pt
2 Centro Recursos Naturais e Ambiente (CERENA), Instituto Superior Técnico, Universidade de Lisboa, Avenida Rovisco Pais 1, 1049-001 Lisboa, Portugal; henrimatos@tecnico.ulisboa.pt
* Correspondence: mciten@isq.pt; Tel.: +351-22-747-1950

**Abstract:** Thermochemical technologies (TCT) enable the promotion of the sustainability and the operation of energy systems, as well as in industrial sites. The thermochemical operations can be applied for energy storage and energy recovery (alternative fuel production from water/wastewater, in particular green hydrogen). TCTs are proven to have a higher energy density and long-term storage compared to standard thermal storage technologies (sensible and latent). Nonetheless, these require further research on their development for the increasing of the technology readiness level (TRL). Since TCTs operate with the same input/outputs streams as other thermal storages (for instance, wastewater and waste heat streams), these may be conceptually analyzed in terms of the integration in Water and Energy Integration System (WEIS). This work is set to review the techno-economic and environmental aspects related to thermochemical energy storage (sorption and reaction-based) and wastewater-to-energy (particular focus on thermochemical water splitting technology), aiming also to assess their potential into WEIS. The exploited technologies are, in general, proved to be suitable to be installed within the conceptualization of WEIS. In the case of TCES technologies, these are proven to be significantly more potential analogues to standard TES technologies on the scope of the conceptualization of WEIS. In the case of energy recovery technologies, although a conceptualization of a pathway to produce usable heat with an input of wastewater, further study has to be performed to fully understand the use of additional fuel in combustion-based processes.

**Keywords:** thermochemical energy storage; water and energy integration systems; energy recovery; thermochemical water splitting; green hydrogen

## 1. Introduction

Thermochemical technologies (TCT) are a set of sorption and reaction-based components and systems having the ultimate objectives of improving the operation and the sustainability of industrial processes and energy systems [1]. For sustainability promotion, TCTs may be implemented with the aims of improving energy use and water use, and reduce the emissions of liquid or gas pollutants, overall promoting the eco-efficiency of industrial units [2,3]. The role of TCTs is, thus, as improvement measures, similar to heat recovery [4] and wastewater treatment [5]. Within system retrofitting research, thermochemical technologies have a fundamental role for water and energy integration systems (WEIS), namely, in the form of alternative thermal energy storage (TES) [6] and wastewater-to-energy (WWtE) [7]. TCTs present overall efficiencies to their counterparts [2], which is substantially notorious in the case of TES [8]. In the case of wastewater-to-energy, TCTs are particularly relevant for green hydrogen production [9]. In this scope, it is to underline the application of thermochemical water splitting (TWS) as an alternative to electrolysis [10].

Thermochemical technologies are essentially applied for management, reduction, recovery, and treatment of material and energy wastes [11]. Within the context of energy

recovery, TCTs are set to be implemented for the promotion of circular economy [12]. Furthermore, the role of these technologies has been proposed for the mitigation of the environment-related impacts of COVID19 pandemic, especially for industrial wastes [11].

The recent EU Energy System Integration Strategy [13,14] pretends the attainment of a net-zero GHG emission-based economy through the promotion of the circular economy perspective on the operation of energy systems. This strategy is itself divided in three pillars: the first pillar (dealing with the energy efficiency and circular economy nexus), the second pillar (dealing with renewable-based electrification), and the third pillar (dealing with alternative low-carbon fuels) [15]. Particular attention shall be applied for the first (which particularly deals with the promotion of waste heat recovery and energy recovery from wastewater) and third pillar (which deals with the promotion of the use of green hydrogen on sectors with more difficult decarbonization) [15].

This work performs a review on the state-of-the-art of thermochemical technologies for energy storage and energy recovery from wastewater. It is set to frame the analysis of the installation of these technologies within water and energy integration systems (WEIS), in follow-up of an ongoing research on this area involving all the aspects associated to the implementation of heat recovery, water treatment and recirculation and eco-efficiency promotion practices in process industry [4,16–19].

## 2. Thermochemical Energy Storage (TCES)

Thermochemical Energy Storage (TCES) comprises a set of technologies which combine both the principles of thermal and chemical energy storage. These may be set to be used in the overall paradigm of the conceptualization of Water and Energy Integration Systems (WEIS) with time-dependent supply and demand levels of water and energy resources. TCES technologies generally present a higher energy storage capacity compared to standard thermal energy storage (in this case, sensible and latent storage technologies) [20]. The characterized technologies have been selected due to their adequacy to the purpose of waste heat recovery in process industry and these are, namely, system-level technologies that may be characterized in three parts: heat source, heat storage units, and heat sinks.

### 2.1. Overview of TCES Potential

Thermochemical energy storage (TCES) technologies are associated to the following advantages [21]: high storage capacity, low heat losses (since energy mat be stored at temperatures near ambient temperature), high storage period, potential of transport at long distance, and high compactness. Nonetheless, these present disadvantages at the level of having high capital costs and being technically complex [22]. In Table 1, these technologies are compared in terms of technical and economic aspects. In Figure 1, the potential of materials set to be used in TCES technologies is compared in terms of energy storage capacity to standard TES materials.

**Table 1.** Comparison of typical values for main performance parameters between different types of thermal storage (adapted from [21]).

| Parameter | Thermal Storage Type | | |
|---|---|---|---|
| | Sensible | Latent | Thermochemical |
| Temperature range (Examples) | Up to 110 °C (Water Tanks) | 20–40 °C (paraffins) 30–80 °C (salt hydrates) | 20–200 °C |
| Storage capacity | 0.2 GJ/m$^3$ | 0.3–0.5 GJ/m$^3$ | 0.5–3 GJ/m$^3$ |
| Lifetime | Long | Limited | Dependable (on reactant degradation and side reactions) |
| Technology status | Commercially available | Partially commercially available | Generally not available |

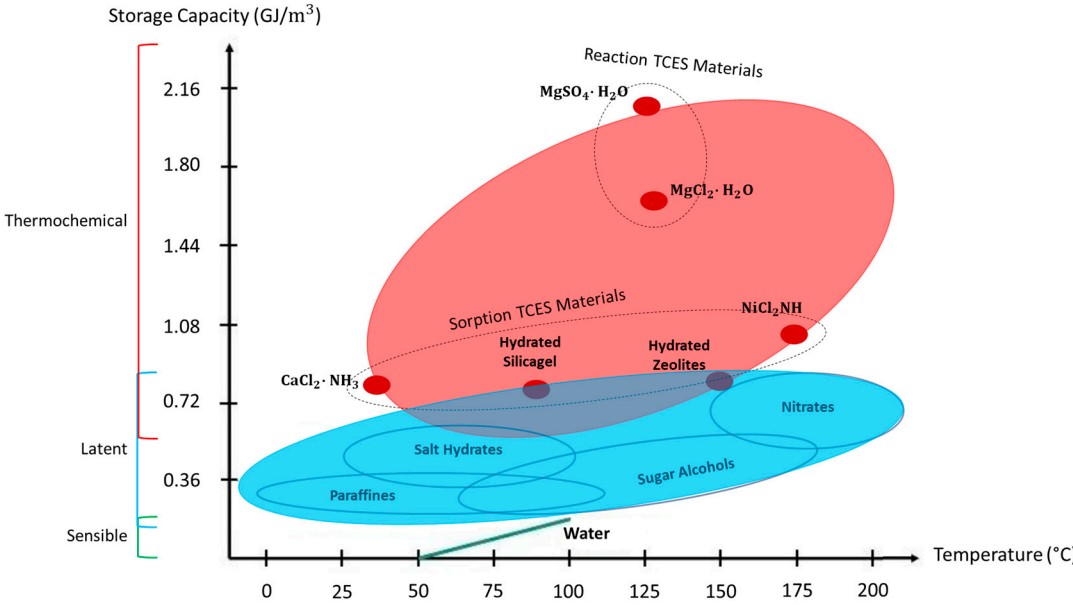

**Figure 1.** Comparison of sensible, latent, and thermochemical storage in terms of temperature range and storage capacity (adapted from [22]).

The data presented in both Table 1 and Figure 1 demonstrate that while the scope of application of TCES is wider than the considered standard TES technologies (wider temperature range and higher energy density), these still suffer from drawbacks in terms of technological maturity. Owing to a relatively low technology readiness level (TRL), TCES technologies do not present the same commercial availability as the standard ones and typical values for the lifetime of these is not deeply understood. The integration of these technologies in the conceptualization of industrial systems whose study is inherent to this work may potentiate the improvement of the TRL of these, and, thus, allow for their constitution into higher potential alternatives to sensible and latent TES.

### 2.2. Description of TCES Technologies

TCES technologies may be either sorption or reaction-based [23]. Sorption technologies may be conceptualized as open or closed systems. In the first, the heat transfer between charging and discharging processes is performed directly between the sorbent and the sorbate. In the latter, the heat transfer occurs in different phases [24]. Reaction-based technologies make use of endothermic and exothermic reactions for the storage of thermal energy (in general, reactions occur in the reactors, where the endothermic reactor corresponds to the heat source and the exothermic reactor to the heat sink) [25]. In general (and as verified by the analysis of Figure 1 above), reaction-based TCES are associated to higher temperature range and energy density than adsorption-based TCES. In Table 2, several TCES technologies are described.

**Table 2.** Description of TCES Technologies and Strategies.

| Technology | Technology Characterization | Operational Conditions | Refs. |
|---|---|---|---|
| Adsorption heat storage (AHS) | It is based on the phenomena of desorption (charging) and adsorption (discharging) of an air stream; The configurations for AHS may be classified into open and closed systems, which are pictorially presented in Figure 2; | Adsorption materials include zeolites (for desorption temperatures up to 180 °C and adsorption temperatures up to 80 °C), aluminophosphates/silico-aluminophosphates (for desorption temperatures of 95–140 °C and adsorption temperatures of 30–40 °C) and metal organic frameworks (for desorption temperatures of 90–140 °C and adsorption temperatures of 30–40 °C). | [26–31] |
| Ammonia-based energy storage | It is based on the reactions of dissociation/synthesis of ammonia ($NH_3$) into/from nitrogen gas ($N_2$) and hydrogen gas ($H_2$) (as described below); It is overall associated to the following advantages: (i) the reaction is single-step and does not require careful control; (ii) the reactants and products are stable at operating temperatures; (iii) the reactants and products are relatively abundant; (iv) possibility for the storage of liquid phase ($NH_3$) and gas phase ($N_2$ and $H_2$) within the same tank due to density differences; The industrial system typically includes two reaction vessels (for dissociation and synthesis), a separation and storage tank and two heat exchangers—Figure 3a. | Operating temperatures overall vary within the range 400–1000 °C. | [32–38] |
| | Reactions | | |
| Haber–Bosch synthesis (Endothermic) | $NH_3 \rightarrow \frac{1}{2}N_2 + \frac{3}{2}H_2$, $\Delta H^0 = +91.8$ kJ/mol | (1) | |
| Calcium-looping energy storage | It is based on the reactions of calcination/carbonation of calcium carbonate ($CaCO_3$) into/from calcium oxide (CaO) and carbon dioxide ($CO_2$) (as described below); The industrial system encompasses three vessels for carbonate, calcium oxide and carbon dioxide—Figure 3b. | Carbonation occurs at about 650 °C, calcination occurs in much more higher temperatures. | [39–43] |
| | Reactions | | |
| Calcination (Endothermic) | $CaCO_3 \rightarrow CaO + CO_2$, $\Delta H^0 = +160 - 172$ kJ/mol | (2) | |
| Metal oxide energy storage | It is based on the reactions of oxidation/reduction of metal oxides (as described below); A typical industrial installation includes the supply of a heat source for the occurrence of reduction reaction and a reactor for the occurrence of the oxidation reaction, as represented in Figure 3c; The reaction enthalpy highly varies for different metal oxides. | The operational temperatures for the occurrence of reaction are set in the range of 700–1400 °C. | [44–50] |
| | Occurring Reaction | | |
| Reduction (Endothermic) | $MO_n \rightarrow MO_{n-\delta} + \frac{\delta}{2}O_2$ | (3) | |

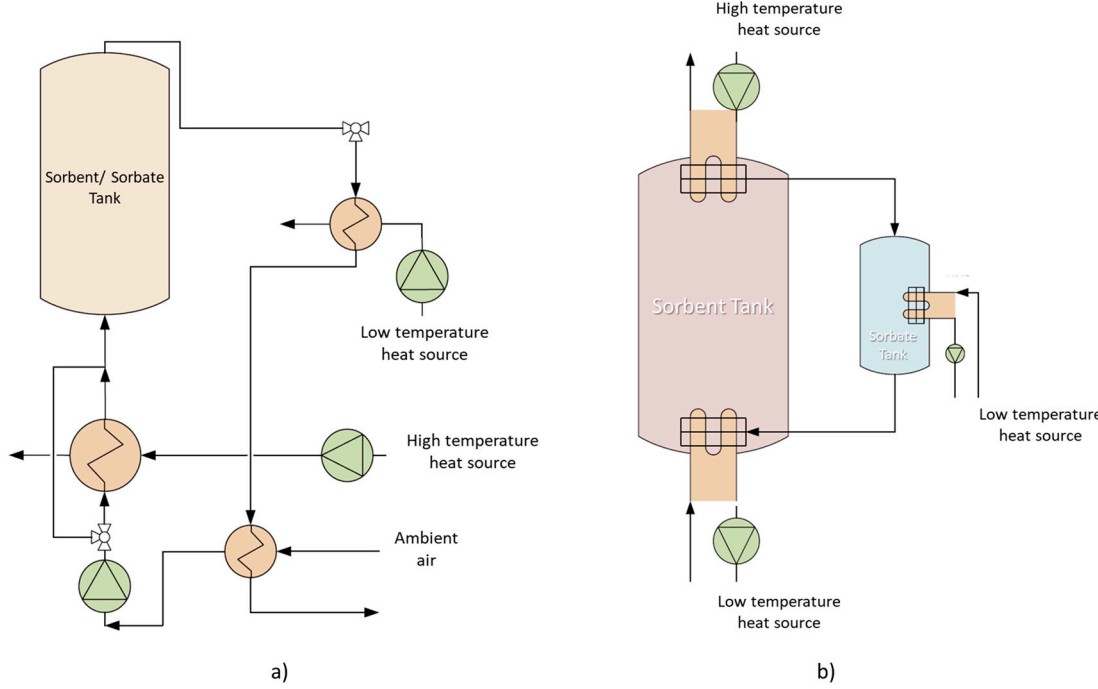

**Figure 2.** Flowsheet for (**a**) Open adsorption heat storage system and (**b**) closed adsorption system (adapted from [29]).

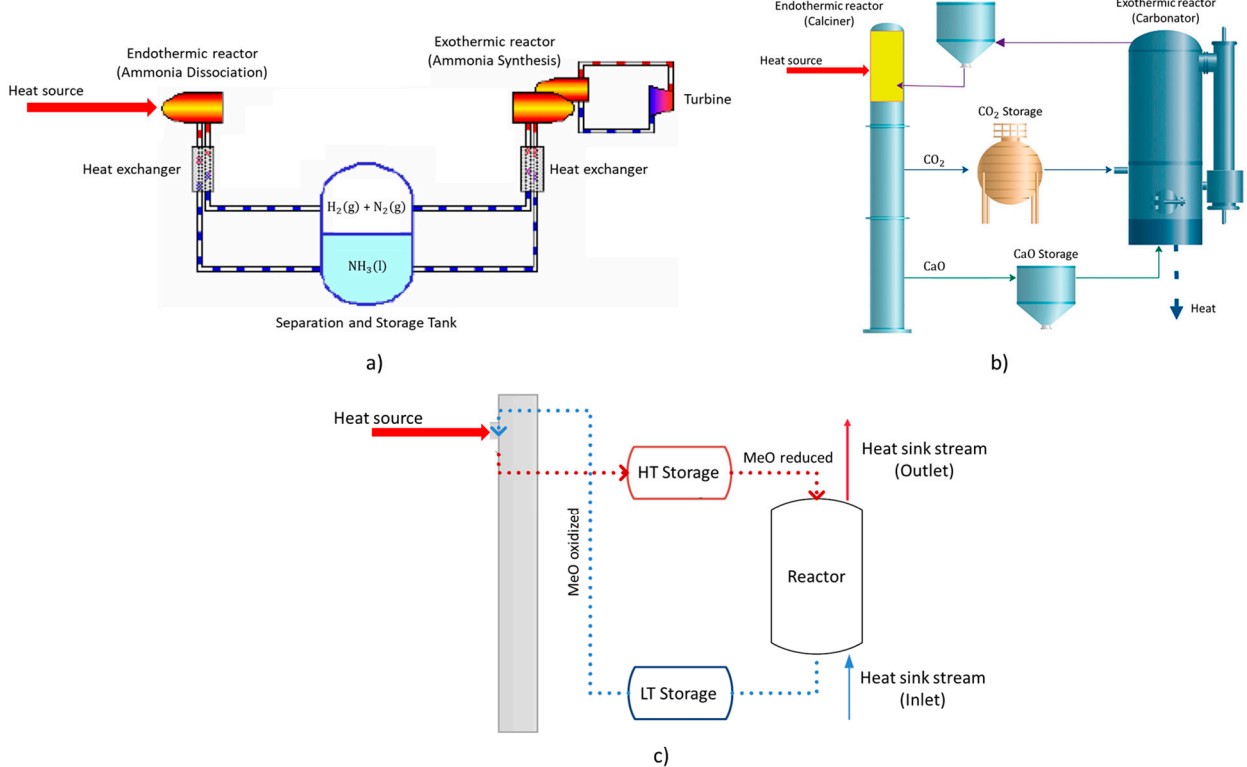

**Figure 3.** Flowsheet for (**a**) Ammonia-based energy storage and (**b**) calcium-looping energy storage (adapted from [39,43]).

## 3. Energy Recovery from Wastewater

Technologies for energy recovery from wastewater may be considered on the overall conceptualization of WEIS. This type of technologies may also be referred as Wastewater-

to-energy (WWtE) technologies and subsists not in the direct use of discharge water and waste heat streams as valorized water and energy inputs, respectively. Rather, these subsist on the use of the liquid water streams or the sludge streams which result as by-products in WWT units for the production of additional quantities of fuels [51]. The selected energy recovery technologies have been chosen based on its adequacy in terms of the produced fuels bearing considering the end-users, namely, combustion-based thermal processes. For instance, as the analyzed processes are natural gas-based, the selected technologies correspond to gaseous fuels production.

### 3.1. Framework of Alternative Fuel Production with Focus on Green Hydrogen

The sludge streams resultant from wastewater treatment may be furtherly valorized to produce additional quantities of fuels (for instance biofuels and synfuels), which may be furtherly used in combustion-based processes (in addition to used primary fuel, such as natural gas) [7]. The particular case of green hydrogen production is a relevant concern on the path to promote innovative low-carbon strategies, with hydrogen being identified as a relevant alternative energy vector [52]. The challenges regarding sustainable hydrogen production include the high cost for hydrogen production technologies and hydrogen distribution within the energy system of a region [53]. In the context of the Portuguese energy system, most recently, the 2020 EN-$H_2$ strategy was approved in the prospect to promote the production and further distribution of green hydrogen. Such strategy prominently aims the creation of an alliance between several Portuguese institutions and enterprises for the development of new technologies, services, and products on this area, which have a period of implementation in the time frame of 2020–2023 [54].

Hydrogen production based on the use of fossil fuels (such as natural gas and coal) represents 95% of the total produced hydrogen, while the remaining 5% correspond to water electrolysis [51,55–57]. Presently, electrolysis is the most mature green hydrogen technology, and the one with highest technology readiness level (TRL). Nevertheless, it is associated to two disadvantages [58]:

- Considerable electricity use (which is already an energy carrier and there is the possibility of additional energy losses by converting it into another energy carrier such as hydrogen);
- Low efficiency of commercial solar panels.

Most recently, thermochemical water splitting (TWS) technologies [59] have been studied as alternative methods to electrolysis for green hydrogen production [9]. This is a set of technologies that (having the supply of a determinate quantity of heat as the driving force [3]) is set for the production of hydrogen from liquid water. The most recent research and development (R&D) efforts have been taken to improve the technology readiness level associated to this set of technologies by improving its economic viability [57]. TWS technologies may be integrated in heat recovery systems, in particular, in ones that use either waste heat or the heat generated by the implementation of solar thermal systems. As such, these may constitute a significant improvement in terms of the whole energy supply and demand chain involving hydrogen as an energy vector. Since these technologies use heat as the driving force, the necessity of additional energy conversion steps for the supply of the required energy input on the waste-to-energy unit is set to be decreased in the context of a whole energy system. In Figure 4, the incorporation of hydrogen within the conceptualization of a sustainable energy systems (encompassing the implementation of TWS and waste heat recovery within a plant) is summarized.

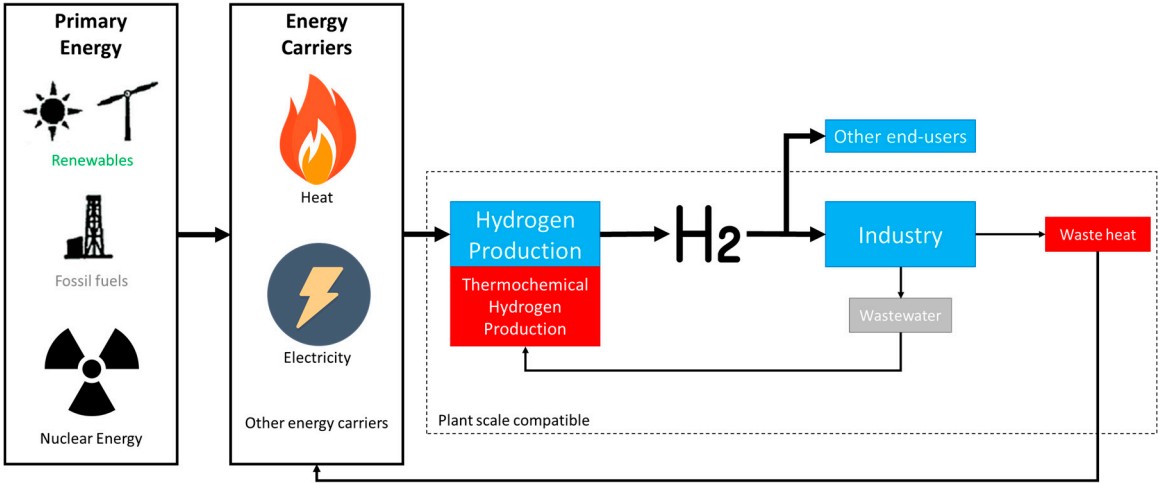

**Figure 4.** Incorporation of hydrogen energy within an energy system with technologies to be analyzed in a plant scale.

### 3.2. Traditional Wastewater-to-Energy Technologies (WWtE)

Traditional wastewater-to-energy (WWtE) technologies with a considerably high technology readiness level include anaerobic digestion (biogas production), gasification (syngas production), and electrolysis (hydrogen production). In the context of green hydrogen production, while the input material stream in an electrolysis and TWS process is only liquid water (which may be the remaining quantity of water present in a sludge stream at the outlet of a wastewater treatment unit) and the main output is hydrogen, the resultant stream from anaerobic digestion and gasification must be further treated to separate hydrogen from other gas components. In Table 3, several traditional WWtE technologies are characterized.

**Table 3.** Characterization of WWtE Technologies.

| Technology | Technology Characterization | Produced Fuel Characterization | Refs. |
|---|---|---|---|
| Anaerobic Digestion | It is a process in which the output is primarily biogas, with a digestate resulting as the by-product; It is prominently applied for the treatment of wastewater streams with a significant load of organic materials, which are considerable prone to biological degradation; The anaerobic digestion process may be integrated in the operation of a WWT unit as represented in Figure 5a. | The produced biogas may be injected in natural gas networks, through the process of separation of carbon dioxide and other contaminants to turn biogas into biomethane. | [60–66] |
| Gasification | It subsists on the partial oxidation of biodegradable material present in wastewater streams for the production of synthesis gas (syngas), as well as a solid fraction of char as by-product; The gasification process may be integrated in the operation of a WWT unit as represented in Figure 5b. | The produced syngas is commonly composed by hydrogen ($H_2$), carbon monoxide (CO), carbon dioxide ($CO_2$) and methane ($CH_4$); The produced syngas is an intermediate in the production of other fuel gases, such as diesel fuel (by Fischer-Tropsch process) and hydrogen (which must be refined for its use in fuel cells). | [67–72] |

**Table 3.** *Cont.*

| Technology | Technology Characterization | Produced Fuel Characterization | Refs. |
|---|---|---|---|
| Electrolysis | It is a process that uses an electric current to produce hydrogen, based on oxidation-reduction reactions; A set of by-products (such as chlorine and sodium hydroxide) may also be generated, as represented in Figure 5c; Several types of electrolysis processes exist, such as: alkaline water electrolysis, solid oxide electrolysis, microbial electrolysis and PEM water electrolysis; | The produced hydrogen may be directly injected into the natural gas fuel supply to combustion-based processes, through processes of production of hydrogen-enriched natural gas (HENG). | [73–82] |

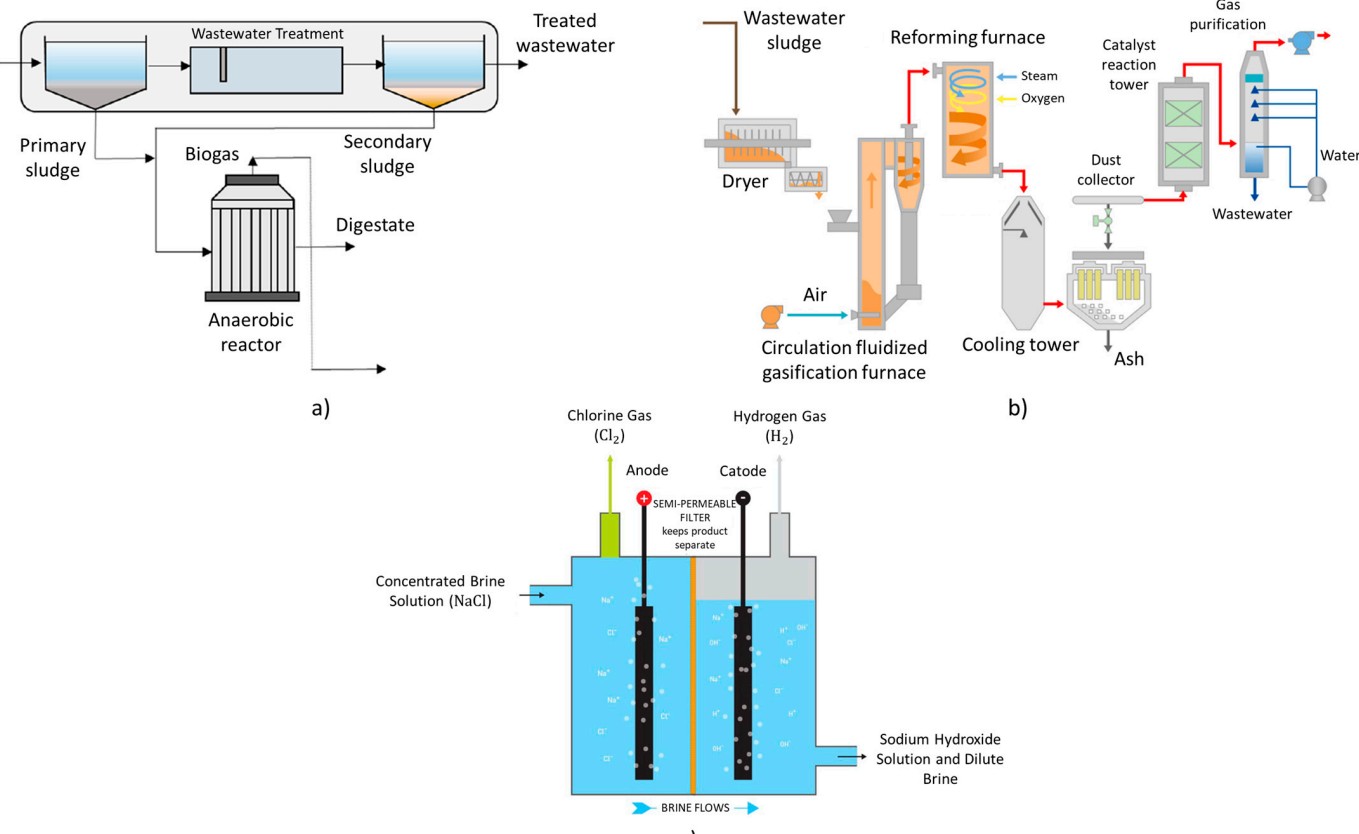

**Figure 5.** Flowsheet representations for WWtE technologies: (**a**) anaerobic digestion, (**b**) gasification, and (**c**) electrolysis (adapted from [62,73,76]).

### 3.3. Thermochemical Water Splitting (TWS)

The implementation of thermochemical water splitting (TWS) technologies sconsists on several thermochemical cycles (whose input is liquid water and whose output is gaseous hydrogen and oxygen) [83]. Each one of these thermal cycles are based on the occurrence of different multistep reactions encompassing two or more reactions, with the overall reaction being water splitting as represented in Equation (4).

$$H_2O \rightarrow H_2 + \frac{1}{2}O_2, \ \Delta H^0 = +241.93 \ kJ/mol \tag{4}$$

The assessment of this technology is based on the analysis of the outputs water streams in a plant, heat source availability (for instance thermal energy through high concentration

of solar radiation or waste heat from nuclear reactors) and cost-related requirements [83]. Thermochemical cycles that are commonly used for the occurrence of thermochemical water splitting include metal oxide cycles, sulfur-iodine (S-I) cycles and iron-chloride (Fe-Cl) cycle. In Table 4, these thermal cycles are characterized.

**Table 4.** Characterization of Thermochemical Water Splitting (TWS) Technologies.

| Technology | Technology Characterization | | Operational Conditions | Refs. |
|---|---|---|---|---|
| Metal oxide cycle | It is a two-step thermal cycle based on the redox reactions of metal oxides (as described below); It presents the following advantages in comparison to the remaining thermal cycles: (i) in terms of input-output streams, wastewater and heat are the only inputs and hydrogen and oxygen are the only outputs; (ii) the produced $H_2$ and $O_2$ are separated in different reactions; (iii) the existence of continuous recycling of reactants and products; (iv) the produced $H_2$ gas is pure; A typical installation encompassing this thermal cycle is represented in Figure 6a; | | Typical metal oxides implemented for this type of thermal cycle are: CdO/Cd, ZnO/Zn, $SnO_2$/SnO, $Mn_2O_3$/MnO, $CeO_2$/$Ce_2O_3$ and $Fe_3O_4$/FeO; The operational temperatures across the cycle are in the range of 900–2000 °C; The reaction enthalpy highly varies for different metal oxides. | [50,59,84–99] |
| | Reactions | | | |
| | Reduction | $MO_n \rightarrow MO_{n-\delta} + \frac{\delta}{2}O_2$ | (5) | |
| | Oxidation | $MO_{n-\delta} + \delta H_2O \rightarrow MO_n + \delta H_2$ | (6) | |
| Sulfur-iodine cycle | It is three-step thermal cycle based on the use of sulfur and iodine components (as described by the reactions below); The advantage of being a significantly high efficiency hydrogen production system, although it as an associated drawback of the involvement of high corrosive sulfuric and iodic acids; A typical installation of this thermal cycle is represented in Figure 6b. | | Reaction (CE7) typically occurs at about 120 °C, (CE8) above 800 °C and (CE9) above 350 °C. | [10,100–104] |
| | Reactions | | | |
| | Sulfuric acid decomposition | $H_2SO_4 \rightarrow SO_2 + H_2O + \frac{1}{2}O_2$, $\Delta H^0 = +186$ kJ/mol | (7) | |
| | Bunsen reaction | $I_2 + SO_2 + 2H_2O \rightarrow 2HI + H_2SO_4$, $\Delta H^0 = -75$ kJ/mol | (8) | |
| | Iodic acid decomposition | $2HI \rightarrow I_2 + H_2$, $\Delta H^0 = +12$ kJ/mol | (9) | |
| Iron-chlorine cycle | It is a four-step thermal cycle based on the use of iron and chlorine components (as described by the reactions below); A typical installation of this thermal cycle is represented in Figure 6c. | | Thermal decomposition occurs at 425 °C, the reverse Deacon reaction and hydrolysis in the range 525–925 °C and chlorination at 125 °C. | [105–107] |
| | Reactions | | | |
| | Thermal Decomposition | $2FeCl_3 \rightarrow 2FeCl_2 + Cl$, $\Delta H^0 = -160.5$ kJ/mol | (10) | |
| | Reverse Deacon Reaction | $Cl_2 + H_2O \rightarrow 2HCl + \frac{1}{2}O_2$, $\Delta H^0 = +59.4$ kJ/mol | (11) | |
| | Chlorination | $Fe_3O_4 + 8HCl \rightarrow FeCl_2 + 2FeCl_3 + 4H_2O$, $\Delta H^0 = -244$ kJ/mol | (12) | |
| | Hydrolysis | $3FeCl_2 + 4H_2O \rightarrow Fe_3O_4 + 6HCl + H_2$, $\Delta H^0 = +156$ kJ/mol | (13) | |

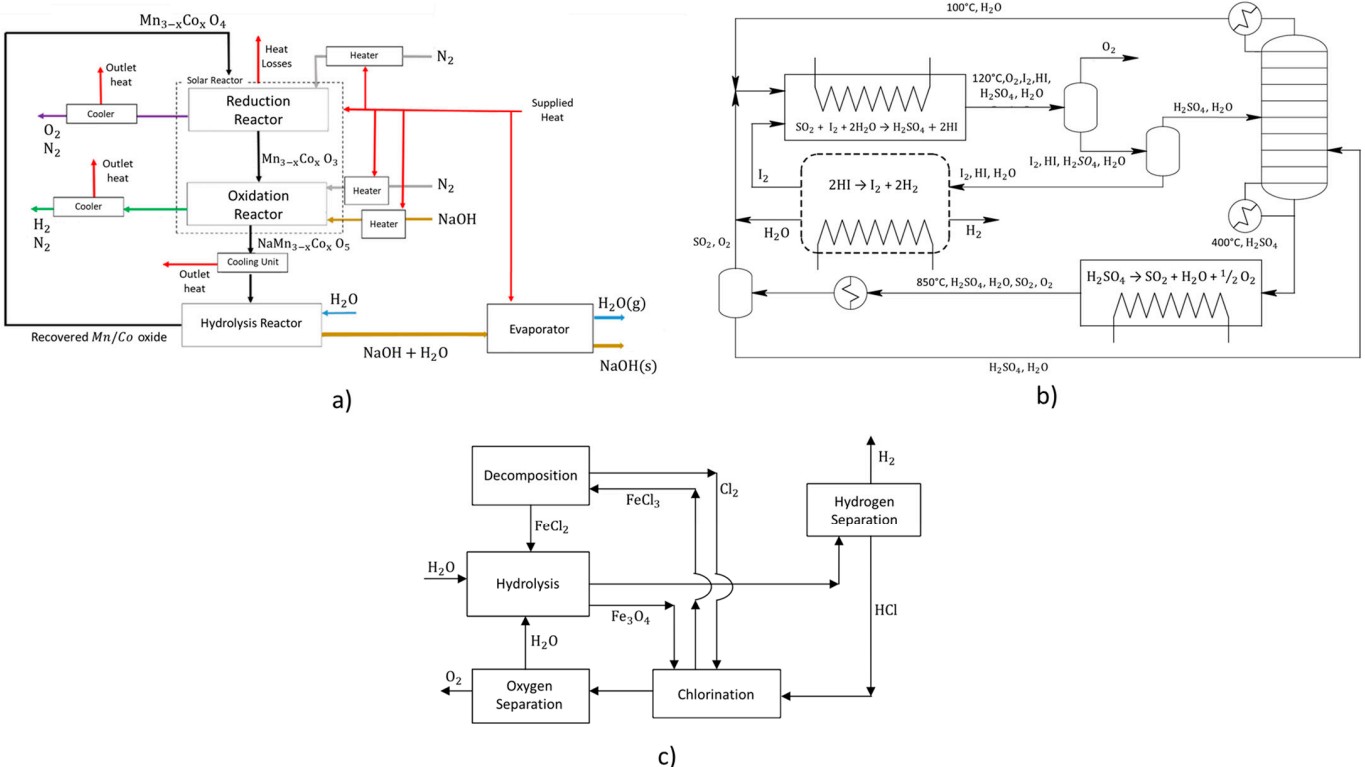

**Figure 6.** Flowsheet for thermochemical water splitting systems: (**a**) Metal oxide cycle, (**b**) Sulfur-iodine cycle, (**c**) Iron-chlorine cycle (adapted from [59,102,107]).

## 4. Thermochemical Technologies for Water and Energy Integration

Thermochemical technologies presented in this work are to be analyzed for integration in Water and Energy Integration Systems (WEIS). The first step for such assessment is the comparison of such technologies to similar ones. Such assessment was already performed in this work, namely, throughout Sections 2 and 3 with the characterization of the technical and economic aspects associated to each technology.

For the full achievement of the objective of proving the superior potential of TCTs within WEIS, it is necessary to analyze the specific benefits in terms of improved potential, namely, water and energy savings and reduced environmental impacts. Table 5 summarizes the existing research on incorporation of thermochemical technology in heat recovery and energy recovery from wastewater.

It is evident that the implementation of thermochemical technologies within the water and energy systems is possible in terms of overall conceptualization (Table 5). In the scope of these studies, these technologies have been successfully exploited in terms of physical phenomena occurrence and equipment sizing. In the case of TCES, the incorporation of these technologies has been proven to be possible within the overall paradigm of waste heat recovery, although the assessment of these as valuable WHR measures has still to be performed for different comparative case-studies (as commonly performed in process integration studies). On the side of WWtE units, further work is still to be developed, namely, in terms of:

- Analysis of the integration of these units for sustainable fuel generation to be used as additional fuel streams in thermal processes;
- Assessment of TWS integration as a (potentially) more efficient form of hydrogen production in comparison to electrolysis;
- Use of waste heat streams as an alternative heat source for TWS instead of solar thermal.

**Table 5.** Characterization of studies for the implementation of Thermochemical Technologies for Water and Energy Integration.

| Aspect | Characterization | Refs. |
|---|---|---|
| Thermochemical Energy Storage for Heat Recovery from Thermal Processes | The functioning of TCES units is analyzed in terms of the supply of variable quantities of thermal energy recovered form waste heat streams; The studies are generally focused on:<br><br>• Analysis of thermal sensitivity;<br>• Analysis of reaction occurrence (in terms of reaction rates and operational temperature and pressure). | [108,109] |
| Use of thermal energy to drive Wastewater-to-energy units | The use of thermal energy (namely the one generated from solar thermal systems) is analyzed for the functioning of WWtE units; These studies are generally focused on:<br><br>• Analysis of the wastewater contaminant concentration on the conditions for fuel generation;<br>• Integration of solar thermal in electrochemical systems for fuel generation. | [110,111] |

In the scope of WEIS conceptualization, TCES technologies are, overall, a highly viable alternative to sensible and latent TES technologies. Their conceptualization shall also consider, on the other hand, the innovative WWtE technological integration within existing WEIS configurations. Figure 7, represents a concept adapted from Oladejo et al. [112] for the integration of a WWtE within a plant considering their thermal processes and wastewater.

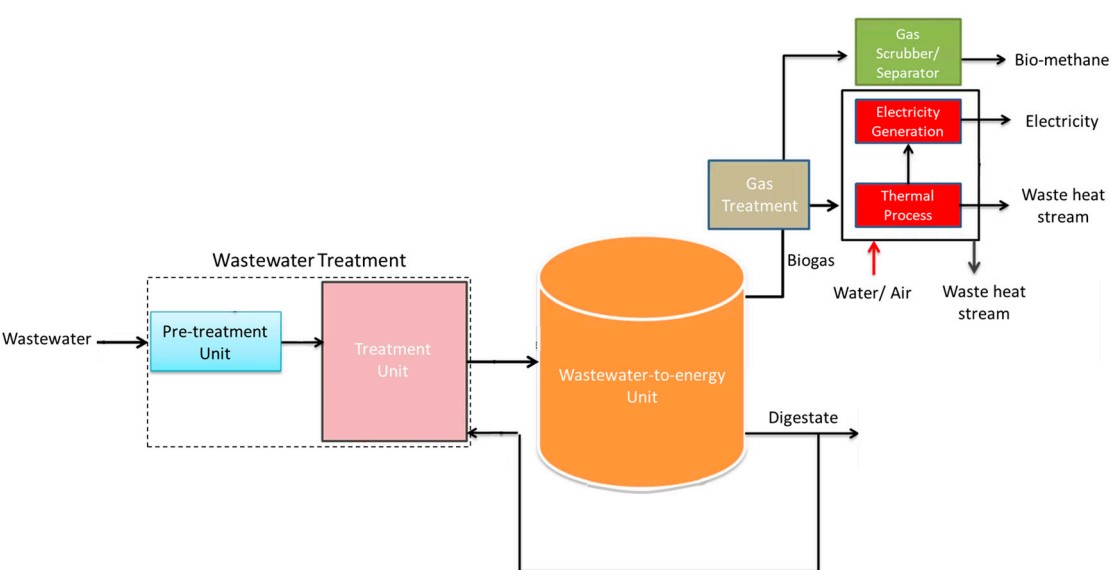

**Figure 7.** Integration of wastewater treatment and wastewater-to-energy units for additional energy generation in thermal processes—WEIS conceptualization (adapted from [112]).

The system represented in Figure 7 enables understanding of the production of additional sustainable fuels from wastewater streams and the allocation of these into thermal process systems (as well as the combustion-based processes within electricity generation systems). Nevertheless, this overall scheme must be analyzed in terms of energy supply and demand requirements (which, in this case, correspond to the WWtE unit and the combustion-based processes, respectively), namely (in the case of gaseous fuels), through a correspondence of the fuel gas to be supplied and the necessary changes to the combustion chambers for the supply of the new gas (for instance, tecno-economic analysis).

## 5. Conclusions

This work is a contribution to the ongoing research on Water and Energy Integration System (WEIS) in Process Industry. The specific aim of this work is the exploitation of the characteristics of several thermochemical technologies (energy storage and wastewater-to-energy), the superior potential of these technologies compared to their counterparts (for instance sensible and latent storages), and the potential to these to be encompassed on the project of WEIS. In relation to thermochemical energy storage, it was verified that:

- Sorption and reaction-based technologies present a significantly higher overall potential compared to standard thermal storage technologies owing to the increased energy storage capacity;
- The approached open and closed system Adsorption Heat Storage (AHS) present an apparatus and operational conditions (lower temperature) that are more adequate to industrial waste heat recovery in comparison to reaction-based technologies (which are commonly set to be installed for the heat source component to be a solar thermal system).

In relation to wastewater-to-energy (WWtE), it was verified that:

- Thermochemical water splitting has a higher operational potential in comparison to the approached traditional technologies (anaerobic digestion, gasification and electrolysis), which is due to the decreased number of steps to produced hydrogen from wastewater (in comparison to anaerobic digestion and gasification) and energy conversion steps (in comparison to electrolysis);
- The required enthalpy to be supplied as the driving force for these technologies is significantly high and requires heat sources which are not compatible with the waste heat potential of industrial sites (they require the supply of thermal energy from nuclear reactors and solar thermal systems instead).

In relation to the overall incorporation of TCTs in the conceptualization of WEIS, it was verified that:

- Thermochemical technologies have been proved to be structurally compatible with the general concept of WEIS, with a small number of studies performing equipment sizing;
- The potential associated to these technologies in terms of environmental benefits is still set to be furtherly calculated (for instance, for the definition of typical and potential values for overall water savings, energy savings and pollutant reduction);
- The existing conceptualization for integration of wastewater treatment and WWtE units for additional energy generation is valid in the scope of the overall concept of WEIS, although it still subsists in a deeper comprehension on energy supply and demand analysis and in terms of the modification of the destination processes for the supply of the new fuels.

In general, as TCES technologies are proved to be suitable as higher potential analogues compared to standard TES technologies. Further work regarding the application of such approaches includes the assessment of their benefits as WHR measures. For energy recovery, further work shall be directed to the study of waste heat-based TWS (one of interest for the conceptualization of WEIS) and its analysis in the integration in each process based on the specific fuel production. Additional study directed to the integration of WWtE within WEIS is being developed in a parallel publication to this paper.

**Author Contributions:** M.C.O., M.I. and H.A.M. performed the literature review; M.C.O. and M.I wrote the paper; M.I. and H.A.M. revised the paper. All authors have read and agreed to the published version of the manuscript.

**Funding:** This work was supported by the European Union's Horizon 2020 research and innovation programmes under grant agreement "No. 810764" and through CERENA under grant UIDB/04028/2020_UIDP/04028/2020.

**Institutional Review Board Statement:** Not applicable.

**Informed Consent Statement:** Not applicable.

**Data Availability Statement:** Not applicable.

**Conflicts of Interest:** The authors declare no conflict of interest.

## Nomenclature

| | |
|---|---|
| AHS | Adsorption heat storage |
| EU | European Union |
| GHG | Greenhouse gases |
| HENG | Hydrogen-enriched natural gas |
| R&D | Research & Development |
| TCES | Thermochemical energy storage |
| TCT | Thermochemical technology |
| TES | Thermal energy storage |
| TRL | Technology readiness level |
| TWS | Thermochemical water splitting |
| WEIS | Water and Energy Integration Systems |
| WHR | Waste heat recovery |
| WWtE | Wastewater-to-energy |

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
