# Peer review of "Review of Thermochemical Technologies for Water and Energy Integration Systems: Energy Storage and Recovery"

_sustainability, doi:10.3390/su14127506_

Round 1

Reviewer 1 Report

In this study, a review study on thermochemical technologies was carried out. This manuscript was found appropriate as it is.

Author Response

Thank you very much for the appreciation!

Reviewer 2 Report

An overview of thermochemical technologies for water and energy integration systems: Energy storage and retrieval is written on 18 pages, applying knowledge from 114 literary sources, which is a sufficient number.

It is necessary to evaluate positively that the article contains nomenclature, reads well and is well oriented in it. It has 5 chapters, 7 pictures, 5 tables. This is a review article which, in my opinion, is well prepared.

The article has a suitable breakdown:

1. Introduction

2. Thermochemical Energy Storage (TCES)

3. Energy Recovery from Wastewater

4. Thermochemical Technologies for Water and Energy Integration

5. Conclusions

The tables are clear. Table 2 provides a convenient overview of thermochemical water storage strategies and technologies, similar to Table 3 of Wastewater-to-energy Technologies and Table 4 of Thermochemical Water Splitting Technologies. Table 5 shows the implementation of Thermochemical Technologies for Water and Energy Integration.

The authors focus directly on national strategies (Portugal), see lines 125-130.

Images are always processed on the basis of information from literary sources but modified by the authors, which is OK.

The conclusion is very concise. The authors point out the benefits of this work. The work is a contribution to the ongoing research of the Water and Energy Integration System (WEIS) in the manufacturing industry. They also indicate that a study on WWtE conceptualization for WEIS is being prepared in the publication in parallel with this work.

I support the publication of the article in the considered journal.

Author Response

(The authors gave the same response as above.)

Reviewer 3 Report

Reviewer comments_sustainability-1756939

The authors of this manuscript review the thermochemical technologies for water and energy integration systems: energy storage and recovery. The results reported in this manuscript deserve to be known by other researchers. But before the publication, several questions should be illustrated more clearly to make the manuscript more readable and meaningful to readers. Detailed comments are as follows:

1.               Abstract and conclusion need to add the significant applications or contributions to body knowledge from this review

2.               Page 2, Line 70: Change these sentences to lowercase “High storage capacity; Low heat losses (since energy mat be stored at temperatures near ambient temperature); High storage period; Potential of transport at long distance and High compactness” and “Nonetheless, it presents disadvantages at the level of being associated to [22]: High capital costs and Being technically complex”.

3.               Standardize the font and size in all Figures. Easier for readers to read

4.               Line 193. “Furtherly, the technical and economic aspects associated with each technology are to be explored, as described in the previous chapters of this work.”…What does this sentence means? Which chapter? Section? Any reference?

5.               Line 272…“Further study on conceptualizing WWtE for WEIS is being developed in a parallel publication to the present work.”.. Present work or present review?

Author Response

Reviewer comments_sustainability-1756939

The authors of this manuscript review the thermochemical technologies for water and energy integration systems: energy storage and recovery. The results reported in this manuscript deserve to be known by other researchers. But before the publication, several questions should be illustrated more clearly to make the manuscript more readable and meaningful to readers. Detailed comments are as follows:

1. Abstract and conclusion need to add the significant applications or contributions to body knowledge from this review

Thank you for the suggestion. The abstract and conclusions include now further information regarding final results. Since this paper is a literature review, the final results include conceptual aspects associated to the implementation of selected technologies within the WEIS. The following has been added:

The characterized technologies were selected because of its adequacy to the purpose of waste heat recovery in the plants of process industry, with these being system-based technologies that may be characterized in three parts: heat source, heat storage units and heat sinks.

And:

In general, as TCES technologies are proved to be suitable as higher potential analogues compared to standard TES technologies. Further work regarding the application of such approaches includes the assessment of their benefits as WHR measures. For energy recovery, further work shall be directed to the study of waste heat-based TWS (one of interest for the conceptualization of WEIS) and its analysis in the integration in each process based on the specific fuel production.

2. Page 2, Line 70: Change these sentences to lowercase “High storage capacity; Low heat losses (since energy mat be stored at temperatures near ambient temperature); High storage period; Potential of transport at long distance and High compactness” and “Nonetheless, it presents disadvantages at the level of being associated to [22]: High capital costs and Being technically complex”.

Thank you for the observation. The upper cases were substituted by lower cases.

3. Standardize the font and size in all Figures. Easier for readers to read

Thank you for the observation. The figures were refurbished to include a standardization of letter types.

4. Line 193. “Furtherly, the technical and economic aspects associated with each technology are to be explored, as described in the previous chapters of this work.”…What does this sentence means? Which chapter? Section? Any reference?

Thank you for the observation. The last two sentences of that paragraph were changed for better understanding by the readers:

Such assessment was already performed in this work, namely throughout sections 2 and 3 with the characterisation of the technical and economic aspects associated to each technology.

5. Line 272…“Further study on conceptualizing WWtE for WEIS is being developed in a parallel publication to the present work.”.. Present work or present review?

Thanks for this.

“this work” has been substituted by “this paper”. The parallel publication is another one we are conceptualizing in addition to this paper that we submitted to Sustainability.

Reviewer 4 Report

The authors review on Thermochemical Technologies for Water and Energy  Integration Systems: Energy Storage and Recovery. Although the topic was interesting, However few things should be incoporated before publication of this article.

Please revised the abstract using a standard format like remove e.g. etc.

add some lines about these technologies, why author choose these Thermochemical technologies.

Mention the full form of any term before using as short form.

Revise the Table 2-4, as author put all the text in one column, in my opinion, all information should be divided into columns.

Mention the advantage and disadvantage of each process involved.

Revise figures as most of them, are in poor quality.

Please take care of some typo and grammetical mistakes thoughout the article.

Precise and brief the conclusion with specific results.

Author Response

The authors review on Thermochemical Technologies for Water and Energy  Integration Systems: Energy Storage and Recovery. Although the topic was interesting, However few things should be incoporated before publication of this article.

Please revised the abstract using a standard format like remove e.g. etc.

Thank you for the observation. All the “e.g.” were substituted by “for instance”.

add some lines about these technologies, why author choose these Thermochemical technologies.

Thank you for the suggestion. Two sets of phrases were added to (respectively) sections 2 and 3 to explain the adequacy of the selected technologies:

The characterised technologies have been selected due to its adequacy to the purpose of waste heat recovery in process industry and these are namely system-level technologies that may be characterised in three parts: heat source, heat storage units and heat sinks.

And:

The selected energy recovery technologies have been chosen based on its adequacy in terms of the produced fuels bearing considering the end-users, namely combustion-based thermal processes. For instance, as the analysed processes are natural gas-based, the selected technologies correspond to gaseous fuels production.

Mention the full form of any term before using as short form.

Thank you for the observation. The manuscript has been checked to verify if the short forms are referred firstly than the full forms.

Revise the Table 2-4, as author put all the text in one column, in my opinion, all information should be divided into columns.

Thank you for the suggestion. Tables 2 – 4 have been amended to include two columns, one for general characterisation of technologies and another for more specific aspects (operational conditions or produced fuel characterization).

Mention the advantage and disadvantage of each process involved.

The advantages and disadvantages have been added for processes/ technologies that have been identified in the literature and that are associated to that specific technology. Technologies not presenting consistent information on advantages/advantages at the literature have not been presented

Revise figures as most of them, are in poor quality.

Thank you for the suggestion. The figures were refurbished to have a greater quality.

Please take care of some typo and grammetical mistakes thoughout the article.

Thank you for the suggestion. Grammar was checked across the article.

Precise and brief the conclusion with specific results.?

Thank you for the suggestion. The abstract and conclusions have been amended to include further information regarding final results. Since this paper corresponds to a literature review, the final results correspond conceptual aspects associated to the implementation of such technologies within the WEIS:

In general, as TCES technologies are proved to be suitable as higher potential analogues compared to standard TES technologies. Further work regarding the application of such approaches includes the assessment of their benefits as WHR measures. For energy recovery, further work shall be directed to the study of waste heat-based TWS (one of interest for the conceptualization of WEIS) and its analysis in the integration in each process based on the specific fuel production.